# Anti-Corrosion Flocking Surface with Enhanced Wettability and Evaporation

**DOI:** 10.3390/ma17164166

**Published:** 2024-08-22

**Authors:** Die Lu, Jing Ni, Zhen Zhang, Kai Feng

**Affiliations:** 1School of Mechanical Engineering, Hangzhou Dianzi University, Hangzhou 310018, China; 221010009@hdu.edu.cn (D.L.); zhangzhen-zz@hdu.edu.cn (Z.Z.); 2School of Mechanical and Electrical Engineering, Lanzhou Jiaotong University, Lanzhou 730070, China; fengkai@lzjtu.edu.cn

**Keywords:** anti-corrosion, electrostatic flocking, wettability, evaporation, electrochemical corrosion

## Abstract

The corrosion protection of tool steel surfaces is of significant importance for ensuring cutting precision and cost savings. However, conventional surface protection measures usually rely on toxic organic solvents, posing threats to the environment and human health. In this regard, an integrated process of laser texturing and electrostatic flocking is introduced as a green anti-corrosion method on a high-speed steel (HSS) surface. Drawing from the principles of textured surface energy barrier reduction and fiber array capillary water evaporation enhancement, a flocking surface with a synergistic optimization of surface wettability and evaporation performance was achieved. Then, contact corrosion tests using 0.1 mol/L of NaCl droplets were performed. Contact angles representing wettability and change in droplet mass representing evaporation properties were collected. The elements and chemical bonds presented on the corroded surfaces were characterized by X-ray photoelectron spectroscopy (XPS). The results revealed that the flocking surface exhibited the lowest degree of corrosion when compared with smooth and textured surfaces. Corrosion resistance of the flocking surface was achieved through the rapid spread and evaporation of droplets, which reduced the reaction time and mitigated electrochemical corrosion. This innovative flocking surface holds promise as an effective treatment in anti-corrosion strategies for cutting tools.

## 1. Introduction

Corrosion of metallic materials is a widespread form of damage to metal facilities, resulting in substantial economic losses [1]. Therefore, it is crucial to pay close attention to corrosion prevention of metal surfaces. Scholars have investigated various anti-corrosion methods, with the following approaches being common: corrosion inhibitors [2], chemical coatings [3], and biological embalming [4]. However, the utilization of toxic substances and organic solvents in metal corrosion protection presents a serious threat to human health, the environment, and ecosystems [5]. Consequently, it is of great significance to develop an environmentally friendly and efficient anti-corrosion technology.

The presence of water droplets/films caused by rain, dew, and salt deliquescence is the main promoter of metallic corrosion in the atmosphere, triggering electrochemical corrosion reactions [6]. Metal facilities, especially those exposed to humid and high-salt coastal areas, are prone to corrosion due to the dissolution of marine chlorides in the moisture layer. These chlorides significantly increase the conductivity of the electrolyte film on the metal and tend to disrupt any protective passivating films [7]. Additionally, common forms of corrosion observed on tool steel surfaces, such as pitting [8] and intergranular corrosion [9], are also prevalent. Therefore, inhibiting the formation of water droplets/films on the metal surface and reducing their contact time are effective means to prevent corrosion.

Superhydrophobic surfaces have been demonstrated to effectively inhibit metal corrosion by allowing droplets to easily roll off the surface. Consequently, the preparation of superhydrophobic surfaces has been extensively studied. Lu and co-workers [10] achieved a corrosion inhibition efficiency of 99.99% by fabricating superhydrophobic aluminum oxide layers on the surface of aluminum through anodization followed by fluorination treatment. However, large-area preparation of hydrophobic surfaces often involves environmentally polluting methods or complex and inefficient processes. Yayoglu et al. [11] discovered that both hydrophobic and hydrophilic surfaces prepared by picosecond laser ablation on polished high-purity magnesium surfaces exhibited similar corrosion rates, which were reduced by 90% compared to polished surfaces. This finding suggested that the preparation of superhydrophobic surfaces was not necessarily required to achieve anti-corrosion. For instance, Yin et al. [12] prepared a hybrid superhydrophobic–hydrophilic surface by femtosecond- laser-induced deposition of polytetrafluoroethylene nanoparticles onto superhydrophobic copper mesh and pristine hydrophilic copper sheet. The surface exhibited enhanced fog collection efficiency compared with uniform hydrophobic or hydrophilic surfaces and an anti-corrosion property. In recent years, these kinds of hybrid wettability surfaces have attracted considerable attention for applications such as directional wetting and water harvesting [13,14,15,16] On a hybrid superhydrophobic/hydrophilic patterned surface, water is transported from the hydrophobic region to the hydrophilic region by wetting gradient forces. Qi and co-workers [17] utilized a simple pulsed UV laser processing technique to create hydrophilic patterned SLIPS (slippery liquid-infused porous surfaces) with superhydrophilic stripes, enabling directional droplet transport. The results showed that the sliding anisotropy of the droplet was influenced by the width and spacing of the hydrophilic stripe, allowing droplet sliding along the designed pattern on the surface. Therefore, hybrid wettability surfaces have the potential to achieve anti-corrosion.

In addition, Wu et al. [18] found that superhydrophilic surface showed a great improvement in evaporation speed. For hydrophilic surfaces, a class of surfaces with rapid wetting and evaporation properties have gained attention, which are prepared using electrostatic flocking technology. Electrostatic flocking is a process that immobilizes electrical charges on the surface of microfibers using a high voltage-connected electrode. It utilizes Coulombic forces to propel these microfibers towards an adhesive-coated substrate, resulting in the formation of a forest of aligned fibers [19]. This surface modification technique has been employed in various applications, including marine antifouling surfaces [20,21], solar-driven interfacial water evaporators [22], biological scaffolds [23,24], electrodes [25], underwater drag-reduction surfaces [26], and other examples. In previous studies, the electrostatic flocking technique has been used to control the wettability of surfaces, specifically hydrophilicity/hydrophobicity. For instance, Xu et al. [27] employed the electrostatic flocking technique and Fe^2+^-catalyzed free radical polymerization to prepare a vertical array of nylon fibers, resulting in zwitterionic electrostatic flocking surfaces (ZEFS). The ZEFS showed strong hydrophilicity, with an underwater oil contact angle of up to 152 ± 2.4° (mean ± standard error). In another study, it was validated that the effect of flocking on surface wettability was related to fiber density, fiber length, and temperature [28]. Flock fibers are typically considered to be small synthetic or natural short fibers, ranging in size from microns to millimeters, with electrically conductive surfaces or finishes. Nylon 6, a widely used engineering plastic and synthetic fiber, possesses favorable properties such as good mechanical strength, impact resistance, melt flow, workability, and excellent corrosion resistance [29]. Consequently, it is frequently employed as a fiber material in electrostatic flocking. Based on the aforementioned research, electrostatic flocking technology has the potential to be used in fabricating hybrid wetting surfaces. These surfaces would allow droplets to penetrate from the hydrophobic metal surface into hydrophilic fiber arrays, thus achieving the goal of metal anti-corrosion.

In this paper, an environmentally friendly approach was utilized to prepare flocking surfaces by combining electrostatic flocking techniques with laser surface texturing. The choice of high-speed steel as the sample material for this study is significant due to its predominant use in the manufacturing of metal cutting tools. Researching the anti-corrosion of high-speed steel holds great practical significance for the manufacturing industry. The wettability and evaporation capacity of different types of surfaces were analyzed. The study also investigated the corrosion of smooth, textured, and flocking HSS surfaces, using a digital microscope to observe and characterize surface corrosion. The elements existing in the sample surface corroded by NaCl solution were characterized via X-ray photoelectron spectroscopy (XPS). Finally, the corrosion resistance mechanism of the flocking surface was explained. This paper aims to propose a new, energy-efficient, and environmentally friendly method for preventing corrosion in metals. The findings have significant implications for the surface protection of metallic materials and hold potential for practical engineering applications, considering the requirements of clean production.

## 2. Materials and Methods

### 2.1. Materials

The nylon fiber (Hexamethylene adipamide) used in the experiment was provided by Dongguan Fuhua Handicraft Co., Ltd., Dongguan, China. Nylon fibers were chosen for their self-adaptive structure and advantageous properties such as abrasion resistance and corrosion resistance. The detailed parameters of the nylon fibers are listed in Table 1. For the matrix material of the sample, high-speed steel (M42) was selected due to its high hardness, good hot hardness, and grindability. Details of the chemical composition of high-speed steel are given in Table 2. The dimensions of the samples were 20 mm × 10 mm × 5 mm. Two types of fluids were used in the experiments: (1) NaCl solution with an initial concentration of 0.1 mol/L, provided by Guangzhou Hewei Pharmaceutical Technology Co., Ltd., Guangzhou, China; (2) Deionized water purchased from Yangzhou Zhongken Food Co., Ltd., Yangzhou, China. The adhesive used to securely fix the nylon fibers in the grooves on the sample surface was supplied by Anda Huatai New Materials Co., Ltd. (Hefei, China).

### 2.2. Preparation of Functional Surface by Electrostatic Flocking

Figure 1 shows the preparation process of the flocking surface. First, as shown in Figure 1a, grooves were fabricated on the sample surface using a laser marking machine (Han’s Laser H20) provided by Han’s Laser Technology Industry Group Co., Ltd., Shenzhen, China, with the specific laser processing parameters listed in Table 3. The surface roughness of the raw sample is Ra 0.1 μm, which is described as a smooth surface in subsequent sections. After laser texturing, sandpaper was used to remove slag from the sample surface. The sample was then cleaned in an ultrasonic cleaner (F-009 SD) provided by Shenzhen Fuyang Technology Group Co., Ltd., Shenzhen, China, for 10 min with deionized water and rinsed with absolute ethanol to ensure surface cleanliness. Then, the sample was placed in a vacuum drying oven (DZF-6050A, Shanghai Kuntian Laboratory Instrument Co., Ltd., Shanghai, China) and dried at a constant temperature of 60 °C for 30 min to obtain the textured surface shown in Figure 1b. Next, adhesive was applied to the dried textured surface using a syringe, as shown in Figure 1c. A squeegee was used to horizontally scrape across the sample surface (Figure 1d), removing excess adhesive and ensuring even distribution of the flocking adhesive within the grooves. The adhesive-coated sample was placed in an electrostatic flocking device (XT-F06), where nylon fibers, 1 mm in length, were vertically implanted into the grooves at a voltage of 100 kV for 5 min (Figure 1e). After flocking, the sample was placed in a vacuum drying oven and dried it at a constant temperature of 60 °C for 24 h to completely solidify the adhesive. Finally, the sample was cleaned in an ultrasonic cleaning machine with absolute ethanol for 3 min to remove any loose fibers from the surface. The resulting topography of the flocking surface is shown in Figure 1f. The density of the flocking surface is 0.079 mg/m^2^.

### 2.3. Experiments and Characterization

The experiment was divided into three parts. The first part involved conducting a wettability test on the sample surfaces. The samples were placed on a contact angle measuring instrument (JC2000D1, Shanghai Zhongchen Digital Technology Equipment Co. LTD, Shanghai, China), and deionized water was dispensed onto the surface in 5 μL increments using a syringe. Images were captured at a frequency of 42 frames per second for a total of 1200 frames per measurement. The contact angle and droplet height were measured to analyze the spreading behavior of droplets on different surfaces. The schematic diagram of the wettability test is shown in Figure 2a. The second part focused on evaluating the surface evaporation performance. The samples were placed on an electronic scale with an accuracy of 0.01 mg (Yingheng Electronic Scale, YHM3004, Shanghai, China), and 50 mg of deionized water was dropped onto each surface using a syringe. The weight on the electronic scale was recorded at 15 min intervals until it stabilized. By comparing the evaporation rates of the droplets on different surfaces, the principles of evaporation and their correlation with surface corrosion were investigated. The third part involved testing the surface corrosion resistance of the samples. A syringe was used to drop 50 μL of 0.1 mol/L of NaCl solution onto each sample surface. NaCl solution was chosen due to its common occurrence as a corrosive agent in natural environments. Corrosion on the sample surface was monitored using a digital microscope (Type: KEYENCE VW-9000, KEYENCE CORPORATION, Osaka, Japan), and images were captured hourly. After the complete evaporation of the NaCl solution, the samples were cleaned with absolute ethanol in an ultrasonic cleaning machine for 5 min. Subsequently, the samples were dried in a vacuum drying oven at a constant temperature of 60 °C for 30 min. Finally, the corrosion spots on the surface were re-examined using the digital microscope and 3D optical profiler. The schematic diagram of the corrosion resistance test is shown in Figure 2b.

## 3. Results and Discussion

### 3.1. Wetting Enhancement of Flocking Surface

As previous research has indicated that the wettability of metal surfaces plays a significant role in corrosion processes, the enhancement of wettability on the HSS surface will be studied. According to Ni et al. [28]’s research, a grooved textured surface demonstrated favorable wetting performance. The optimal parameters for the width, spacing, and depth of the grooves were found to be 0.6 mm, 1.75 mm, and 0.5 mm, respectively. Based on these findings, textured surfaces and flocking surfaces with different parameters were designed and prepared. Wettability tests were performed on each surface to optimize the texture parameters and achieve the optimal surface wettability. Figure 3a illustrates four different textured surfaces labeled T1, T2, T3, and T4. Figure 3b illustrates four flocking surfaces with different parameters corresponding to the textured surfaces, labeled FT1, FT2, FT3, and FT4. The specific groove parameter for the surfaces is presented in Figure 3c. All grooves have consistent dimensions of width, depth, and spacing: 0.6 mm, 0.5 mm, and 1.75 mm, respectively. Grooves of T2, T3, and T4 are evenly spaced 1 mm apart vertically. The lengths of the individual grooves for T1, T2, T3, and T4 are 12 mm, 5.5 mm, 3.3 mm, and 2.25 mm, respectively. Additionally, a micrograph of the flocking surface was obtained using a 3D optical profiler (S neox 090, Shanghai Sensofar Optical Precision Instrument Co., Ltd., Shanghai, China), as shown in Figure 3d.

Wettability tests were conducted on the HSS samples with different parameters to investigate their wettability properties. A 5 μL droplet of deionized water was placed on each sample surface, and the spreading of the droplet was observed to assess the wettability of the surfaces. During the preliminary experiments, it was observed that the droplets rapidly spread within 0.2 s upon contacting with the flocking surface, and their height decreased quickly. After 0.2 s, the spreading rate of the droplets decreased significantly. To facilitate observation and to efficiently extract useful information from the images, images were captured at intervals of 0.1 s starting from 0 s, and at intervals of 2 s starting from 1 s.

For the wettability experiments on textured surfaces, three samples were prepared for each one as part of a repeatability experiment. The average contact angle was calculated from the measurements. The direction of the contact angle measurement is parallel to the groove direction (longitudinal) of the sample surface. Figure 4a illustrates a random group in the repeatability experiment. The average contact angle is shown in Figure 4b. For the non-flocking surfaces, the contact angles of the four textured surfaces were initially very close when the droplet first landed on the surface. Over time, the contact angle on the T1 surface remained constant, while on the T2, T3, and T4 surfaces it decreased. Among them, the contact angle on the T2 surface stabilized at 100° after 0.1 s, while on the T3 and T4 surfaces it remained stable at 77.5° and 74°, respectively, after 3 s. When the contact angles on all surfaces reached stability, the numerical order of the contact angles for the four textured surfaces was T1 > T2 > T3 > T4. The occurrence of boundary slip in lubricated contacts depends on the shear stress at the solid/liquid boundary reaching a yield value. From a molecular perspective, the transition from a no-slip condition to a slip condition indicates that the liquid molecules have gained enough energy to overcome the interfacial energy barrier. The longer the contact distance between the droplet and the groove edge, the more challenging it is for the droplet to transition from a no-slip state to a slip state, resulting in a higher interfacial energy barrier from the Cassie state to the Wenzel state [30,31]. Since the groove on the T1 surface was the longest and had the longest contact distance with the droplet in the longitudinal direction, it had the highest interfacial energy barrier. As a result, it was difficult for the droplets to spread transversely on the T1 surface, and the contact angle remained large throughout. For surfaces T2, T3, and T4, the contact distance between the groove and the droplet decreased successively, leading to a reduction in the interfacial energy barrier and thus resulting in a sequential decrease stable contact angle. As shown in Figure 4c, the stable contact angle was measured parallelly and perpendicularly to the grooves. In the parallel direction, the stable contact angles for the T1, T2, T3, and T4 surfaces were 137°, 100°, 77.5°, and 74°, respectively. In the perpendicular direction, the stable contact angles observed were 87°, 81°, 66.5°, and 60.5°, respectively. The order of contact angle values was T1 > T2 > T3 > T4. Among them, the largest difference in contact angles between the parallel and perpendicular directions was observed on the T1 surface (50°), followed by the T2 surface (19°), T4 surface (13.5°), and T3 surface (11°). This showed that the T1 surface, with the longest groove length, exhibited the largest difference in wettability between the parallel and perpendicular directions, while the smallest difference was observed on the T3 surface.

For the wettability experiment on flocking surfaces, three samples were prepared for each flocking surface to ensure repeatability. The spreading of droplets on the flocking surface is shown in Figure 5a. Considering the complete wetting observed in pre-tests, there were challenges in measuring the surface contact angle on the FT1 and FT2 surfaces, so droplet height was chosen as a suitable parameter to evaluate the variances in wettability among different samples. The average droplet heights were calculated as depicted in Figure 5b. Upon the initial landing of the droplet on the FT1 surface, the droplet’s height (h) measured 1.75 mm. Within 3 s, this height reduced to 0, an indication of the surface’s complete wetting. On the other three surfaces (FT2, FT3, and FT4), the droplet height did not reach 0. After 3 s, the droplet heights on the FT2 and FT3 surfaces stabilized at 0.67 mm and 1.01 mm, respectively, while on the FT4 surface, the droplet height stabilized at 1.43 mm after 1 s. When the droplet height on all surfaces remained constant, the final droplet height sequence for the four flocking surfaces was FT4 > FT3 > FT2 > FT1. These results indicate that the flocking surface with a connected texture (FT1) exhibits the best wetting effect. Due to the presence of nylon fibers, the capillary force required for droplet adhesion increased, causing the droplet to be rapidly drawn into the fiber gaps. The grooves and fiber gaps in the flocking surface played a role in storing the droplets. When the grooves and fibers near the droplet reached their maximum liquid adsorption capacity, the droplet stopped spreading. FT1, with the longest connection length inside the groove, had the lowest interfacial energy barrier parallel to the groove. This resulted in the fastest spreading rate and the ability to attract and retain the most liquid. In summary, the different energy barriers caused by the length of the texture affected the spreading of the droplets on the surface along the direction perpendicular to the grooves, resulting in the final contact angles being T1 > T2 > T3 > T4. Furthermore, the contact length between the flocking grooves and the droplets affected the volume of liquid stored, leading to height variations of the droplets on the textured surface, with the final heights being FT4 > FT3 > FT2 > FT1.

### 3.2. Evaporation Enhancement of Flocking Surface

As it is well-known that the corrosion level of a metal surface is directly influenced by the duration of the corrosion reaction, with longer droplet dwell times leading to more pronounced corrosion, the evaporative properties of droplets on various HSS surfaces will be studied. To investigate the behavior of droplets on sample surfaces and to provide valuable insights for corrosion resistance of metal surfaces, a comparative study of droplet evaporation across different surface types was performed. The test surfaces were smooth surfaces, T1 surfaces, and FT1 surfaces, as FT1 showed the best wettability according to the results mentioned in Section 3.1. Deionized water (50 mg) was dropped onto the smooth surface, textured surface, and flocking surface. The masses of the samples were measured using an electronic balance at 15 min intervals. Five sets of experiments were conducted simultaneously under identical working conditions for reliable results. The change in mass (subtracting the initial mass) was recorded after each weighing, and the average of the five sets of data was calculated. The experiments were conducted in an unventilated clean room with a room temperature of 26 °C and an air humidity of 44%.

As shown in the line diagram of Figure 6, the droplet exhibits the shortest retention time on the flocking surface, evaporating completely within 105 min. The second is the textured surface with a retention time of 120 min. The smooth surface shows the longest droplet retention time, taking 135 min to evaporate. The evaporation rate of the flocking surface was 28.57% higher than that of the smooth surface and 14.29% higher than that of the textured surface.

Figure 7 provides a schematic representation of droplet evaporation on the three different surfaces. In the initial period, as shown in Figure 7a, the poor wettability of the smooth surface caused water droplets to accumulate with a significant thickness. The small surface area in contact with the air resulted in a slow evaporation rate. In the late stage of evaporation (Figure 7b), the droplet thickness reduced, leading to a decreased surface tension and accelerated evaporation [32]. Figure 7g shows the evaporating surface of a droplet on the smooth surface. On the textured surface, the droplet mass decreased rapidly initially, but when approaching the drying stage, the rate of mass decline became slower compared to the smooth surface. This behavior can be attributed to the water droplets quickly spreading into the grooves initially, increasing the surface area in contact with air and facilitating faster evaporation (Figure 7c,d). However, it entered a period of slow evaporation in the late stage, due to the small amount of liquid remaining in the grooves and limited exposure to air. Figure 7h shows the change in evaporating surface.

As shown in Figure 7e, during the initial stage, the height of the droplet exceeds that of the fibers protruding from the surface. Due to the excellent wettability of the flocking surface, water droplets on the flocking surface would spread towards the grooves and gaps between the fibers, maximizing the evaporation surface area, as depicted in Figure 7f. Figure 7i shows the change in the evaporating surface of the flocking surface from Ⅰ to Ⅱ. As illustrated in Figure 8, water on the flocking surface was attracted to the spaces between the nylon fibers due to the capillary force. The presence of hydrophilic amide groups in the nylon fibers facilitated the formation of hydrogen bonds with water molecules, resulting in good hydrophilicity and spontaneous attraction of fibers to water droplets. As the water droplet spread among the fibers, the surface area increased, allowing for faster evaporation. Moreover, the water within the nylon fiber array remained in a capillary water state. During evaporation, steam was formed in clusters consisting of a few to tens of molecules. It required less energy for these water clusters to evaporate compared to individual molecules in bulk water, leading to an accelerated evaporation rate [22,33]. These findings highlighted the distinct evaporation properties of the droplets on the three surfaces, with the flocking surface exhibiting the highest evaporation rate, followed by the textured surface and the smooth surface. Combined with the results in Section 3.1, it is indicated that good wettability enhances surface evaporation.

### 3.3. Corrosion Resistance of Flocking Surface

#### 3.3.1. Corrosion Process

To evaluate the corrosion resistance of the flocking surface, a comparison test was performed under the same working conditions. Nine HSS samples were prepared: three smooth surfaces, three textured surfaces, and three flocking surfaces. Following this, 50 μL of 0.1 mol/L of NaCl solution was, respectively, dripped onto the smooth surface, textured surface, and flocking surface. Three samples were prepared for each surface to ensure repeatability. Images of surface corrosion were captured every hour using a digital microscope at a magnification of 20×. After a duration of 2 h, the NaCl solution on the sample surface was completely evaporated. The cleaned and dried samples were observed using a digital microscope at a magnification of 200×.

Figure 9 illustrates the corrosion of different surfaces when exposed to NaCl solution. As depicted in Figure 9(a1–a3), the smooth surface exhibited significant corrosion, with visible corrosion spots and pitting pits appearing after just 1 h of corrosion. After 2 h, a large number of corrosion spots were evident. Following ultrasonic cleaning, the stains generated during the corrosion process were removed, but the severely corroded zone still contained residual corrosion spots (Figure 9(d1,d2)). In comparison, textured surfaces (Figure 9(b1–b3)) exhibited less corrosion than smooth surfaces. Corrosion started at the edge of the droplet and gradually expanded inward. The surface did not exhibit a large number of corrosion spots until the droplet had completely evaporated, but some residual corrosion stains remained (Figure 9(e1,e2)) after ultrasonic cleaning. In contrast, the flocking surface (Figure 9(c1–c3)) showed superior corrosion resistance. After 1 h of exposure to the NaCl solution, the droplets on the flocking surface remained clear, with no corrosion observed at the edges of the droplets. Only a small amount of corrosion appeared on the surface after 2 h. Following ultrasonic cleaning, only minimal corrosion stains remained (Figure 9(f1,f2)). Additionally, to facilitate a clearer observation of corrosion on the surfaces, a 3D optical profiler was used to capture images. As the image shows, large and deep pitting pits appeared on the smooth surface (Figure 9g). The maximum pitting depth is 6.67 microns. On the textured surface (Figure 9h), some small and shallow corrosion pits appeared, indicating a lower corrosion degree. As shown in Figure 9i, no obvious pitting corrosion was observed on the flocking surface, indicating the lowest corrosion degree. Thus, the flocking surface demonstrated the best corrosion resistance among the tested surfaces. 

The NaCl solution forms oxygen concentration cells on the metal surface, leading to electrochemical corrosion. The principle of the corrosion process is illustrated in Figure 10. On a smooth surface, droplets tend to have a larger thickness (Figure 10a) due to poor surface wetting. As a result, oxygen has limited access to the bottom of the droplet near the surface, leading to a low oxygen concentration in that region. In the oxygen-deprived region near the HSS surface, the natural potential (non-equilibrium potential) is low, forming the anode of a corrosion galvanic cell. In turn, the surface of the droplet, in direct contact with the air, has a higher concentration of oxygen, creating the cathode of the corrosion galvanic cell. The anodic dissolution process causes the metal to corrode. Since smooth metal surfaces generally exhibit poor wettability, indicated by a large contact angle, the difference in the concentration of oxygen within the droplet is the most pronounced. Consequently, the anodic dissolution rate on smooth surfaces is significantly larger than that on other surfaces. Due to its unfavorable wettability and large contact angle, the smooth HSS surface experiences the most substantial corrosion due to the significant difference in oxygen concentration within the droplet. When the oxygen content in the droplet is low, Fe(OH)_2_ can form as an intermediate product during the corrosion process of Fe in a NaCl solution. This occurs through the following reaction: Fe + 2H_2_O = Fe(OH)_2_ + H_2_(1)

Fe(OH)_2_ is a soluble precipitate that can further undergo oxidation to Fe(OH)_3_. As the reaction progresses (Figure 10b), a greater amount of oxygen reaches the metal surface. The produced Fe(OH)_2_ is further oxidized to Fe(OH)_3_, which then precipitates on the metal surface. This precipitate results in the formation of a passivation film. The reaction equation for this process is as follows:4Fe + 3O_2_ + 6H_2_O = 4Fe(OH)_3_(2)

On the surface covered by the passivation film, the metal is protected from corrosion because the direct contact between the metal surface and the corrosive environment is limited. However, passivation film defects, such as scratches or discontinuities, can lead to pitting corrosion. Pitting corrosion is facilitated by the presence of aggressive anions, typically halides like Cl^−^ ions, in the solution [34]. At the defect of the passivation film, the bare metal is exposed and comes into direct contact with the NaCl solution. In this localized area, the active Cl^−^ anions combine with the Fe^3+^ cation present in the passivation film, forming a soluble compound.

Corrosion spots form and gradually develop into corrosion holes. Inside the corrosion hole (as shown in Figure 10b), the exposed metal surface exhibits high activity and low potential, constituting an anode. On the other hand, the surface outside the corrosion hole, protected by the passivation film, has low activity and high potential, forming a cathode. This configuration creates a small corrosion galvanic cell between the inside and outside of the corrosion hole. The metal at the anode inside the corrosion hole undergoes rapid dissolution, generating a large amount of Fe^2+^. Since dissolved oxygen finds it difficult to enter the corrosion hole, significant amounts of Cl^−^ ions migrate into the hole to maintain charge balance. When Cl^−^ ions solely exist and migrate into the occluded area through the rust layer, the progression of the occluded area follows an acidification autocatalytic process, thus lowering the pH value of the solution and increasing its acidity [35].

The increased acidity in the corrosive pores leads to a continued acceleration of the corrosion, resulting in an expansion of the depth and width of the pores, eventually leaving visible pitting pits in the surface. In Figure 10c, after the droplet evaporates, the residual Fe(OH)₃ on the sample surface progressively dehydrates in the air, resulting in the formation of Fe_2_O_3_. The reaction equation is as follows:4Fe(OH)_3_ = 2Fe_2_O_3_ + 6H_2_O(3)

The surface ultimately develops a reddish-brown rust. The reddish-brown color is characteristic of the iron oxide formed during the corrosion process and serves as a visual indication of the extent of the corrosion damage to the metal surface. 

The whole process can be reduced to the following synthesis equation: 4Fe + 3O_2_ + 6H_2_O = 2Fe_2_O_3_ + 6H_2_O(4)

With the formation of Fe_2_O_3_, fresh metal surfaces are exposed. These newly exposed surfaces are more reactive and susceptible to further corrosion processes, including the initiation of new pits.

On a textured surface, as illustrated in Figure 10d, the presence of grooves allows for better wetting compared to a smooth surface and the droplet thickness is smaller. As the thickness of the droplet decreases, the amount of dissolved oxygen at the center of the droplet increases and the oxygen concentration difference in the vertical direction decreases. In addition, the evaporation phenomenon leads to a hydrodynamic flow and liquid will flow toward the droplet edge to replenish the excess peripheral evaporation losses, resulting in an enrichment of the NaCl. The concentration of NaCl solution at the edge of the droplet increases. This reduces the solubility of oxygen in the concentrated NaCl solution near the contact line [36]. Consequently, the oxygen concentration at the center of the droplet is higher than at the edges, leading to an inversion of the droplet corrosion model depicted in Figure 10a. In this revised model, the anode of the corrosion galvanic cell is positioned at the edge of the droplet, while the cathode is located at the center, initiating corrosion from the edges inward.

When a NaCl solution is dropped on a flocking surface, capillary forces between the nylon fibers in the grooves and the hydrophilicity of the fibers cause the droplets to spread rapidly across the surface. Most of the droplets are attracted into the grooves and absorbed within the gaps of the nylon fibers, leaving only a small amount of salt solution on the metal surface. As depicted in Figure 10e, on the metal surface, the droplet thickness is small, making it easier for oxygen to reach and resulting in a high oxygen concentration. Conversely, the thickness of the droplets at the nylon fibers is larger, making it more difficult for oxygen to enter and resulting in a lower oxygen concentration. Thus, it becomes challenging to establish a corrosion galvanic cell, and the metal surface is less prone to corrosion. Moreover, the NaCl solution at the edge of the droplet evaporates so quickly that the reaction time with the metal surface is short. Only areas with thicker droplets will retain some small spots caused by electrochemical corrosion, while the rest of the surface remains relatively unaffected. 

#### 3.3.2. XPS 

In order to verify the above principle, X-ray photoelectron spectroscopy (XPS) was used to analyze the surface composition of the sample corroded by 0.1 mol/L of NaCl solution. The test area of each sample is 0.25 mm^2^. The XPS spectra were fitted using the Avantage software (Version number: 5.9931.0.6755), with the charge correction performed using the C1s peak at the binding energy of 284.8 eV. Figure 11 shows the typical XPS spectra for the smooth surface, textured surface, and flocking surface. It was demonstrated that the peak positions of Fe 2p were influenced by the ionic state of Fe, and the satellite peak positions of the Fe 2p peaks were also sensitive to the oxidation state [37]. Figure 11a–c shows bonding energies at 711.9 eV and 724.7 eV, which corresponded to Fe 2p2/3 and Fe 2p1/2, respectively [38]. A satellite peak at 719.5 eV was also observed, suggesting the presence of Fe_2_O_3_ after surface oxidation. The fitting analysis identified the species Fe^3+^, Fe^2+^, and Fe, with signals at binding energies of 711.9/724.7 eV, 710.2/723.0 eV, and 706.7/719.5 eV, respectively.

The relative contents of Fe^3+^, Fe^2+^, and Fe on the corroded sample surface are presented in Figure 12a. On the flocking surface, the lowest relative content of Fe^3+^ was observed (37.71%), while the textured surface had 53.52%, and the smooth surface had the highest content at 78.37%. The relative content of Fe^2+^ was highest on the flocking surface (49.28%), followed by the textured surface (35.10%), with the lowest on the smooth surface (20.21%). Similarly, the relative content of Fe was highest on the flocking surface (13.01%), followed by the textured surface (11.38%), with the lowest on the smooth surface (1.42%). These results indicated varying degrees of Fe oxidation on the sample surface after corroding with NaCl solution, and the flocking surface showed the lowest. The high wettability of the flocking surface leaded to a wider spread of droplets, resulting in smaller oxygen concentration differences and lower electrochemical reaction rates. Additionally, the rapid evaporation of the droplets meant that oxygen acted on the surface for a shorter duration, resulting in higher Fe relative content and lower Fe^3+^ relative content at the flocking surface. Figure 11d–f reveals strong O 1s peaks at 529.5 eV and 531.5 eV, indicating the presence of Fe_2_O_3_ and Na_2_CO_3_ on the surface. The relative contents of O element in Fe_2_O_3_ and Na_2_CO_3_ are shown in Figure 12b, with the lowest relative content of O in Fe_2_O_3_ observed on the flocking surface (35.27%). This indicated a lower degree of oxidation on the flocking surface, further confirming its anti-corrosion effect. The results of the XPS analysis were in agreement with the findings presented in Figure 10, thereby providing additional evidence for the robust corrosion resistance of the flocking surface.

## 4. Conclusions

The study aimed to improve the corrosion resistance of HSS surfaces by using lasers to create groove-type textures and by implanting nylon fibers into the grooves through electrostatic flocking equipment. Wettability, evaporation, and anti-corrosion capabilities on different surfaces were evaluated. The following are the main conclusions of the study:The sample surfaces with different texture shapes showed varying contact angles (CAs): T1 > T2 > T3 > T4. The T1 surface exhibited the greatest difference in wettability between parallel and perpendicular directions due to its highest transverse interface barrier. The FT1 surface demonstrated the best droplet spreading performance, which achieved complete wetting within 3 s, as the grooves and fiber gaps facilitated droplet spread and storage. Surfaces with longer grooves had lower energy barriers parallel to the grooves, attracting and retaining more droplets;The flocking surface exhibited enhanced evaporation ability compared to the smooth and textured surfaces. The evaporation rate on the flocking surface was increased by 28.57% compared to the smooth surface and 14.29% compared to the textured surface. This was attributed to the expansion of the water droplet evaporation region on the flocking surface and the capillary state of the water within the nylon array, enabling evaporation in the form of water clusters;The flocking surface demonstrated superior anti-corrosion properties compared to the smooth and textured surfaces under similar working conditions. The XPS analysis confirms the lowest degree of oxidation at the flocking surface. This was attributed to the rapid evaporation of the droplets at the flocking surface, which reduced the electrochemical reaction time at the metal surface. Additionally, the capillary force and hydrophilicity among the nylon fibers in the grooves facilitated the aggregation of liquid on the nylon fibers, inhibiting the formation of corrosion galvanic cells.

## Figures and Tables

**Figure 1 materials-17-04166-f001:**
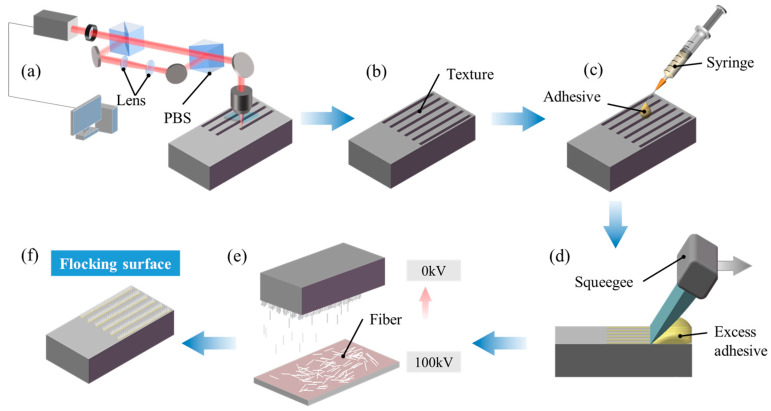
Electrostatic flocking process on the sample surface. (**a**) Schematic diagram of laser texturing. (**b**) Textured surface. (**c**) Drop adhesive. (**d**) Scrape off excess adhesive. (**e**) Implant fibers. (**f**) Prepared flocking surface. (Blue arrows) Process sequential orientation. (Gray arrow) Moving direction of squeezer. (Pink arrow) Fiber implantation direction.

**Figure 2 materials-17-04166-f002:**
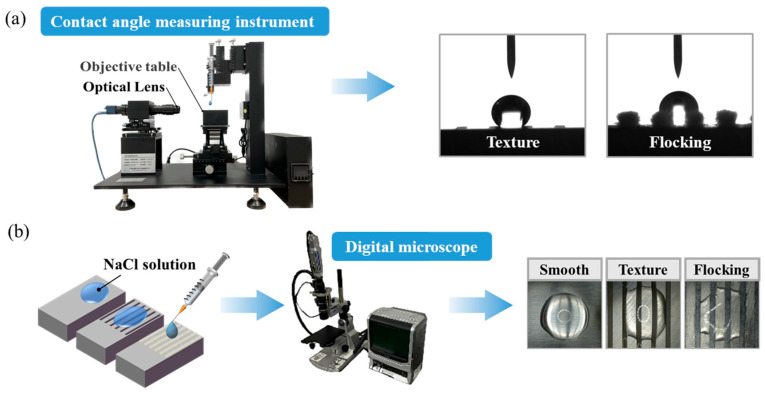
Experimental facilities and processes. (**a**) Wettability test. (**b**) Corrosion resistance test. (Arrows) Sequence of test operation and observation.

**Figure 3 materials-17-04166-f003:**
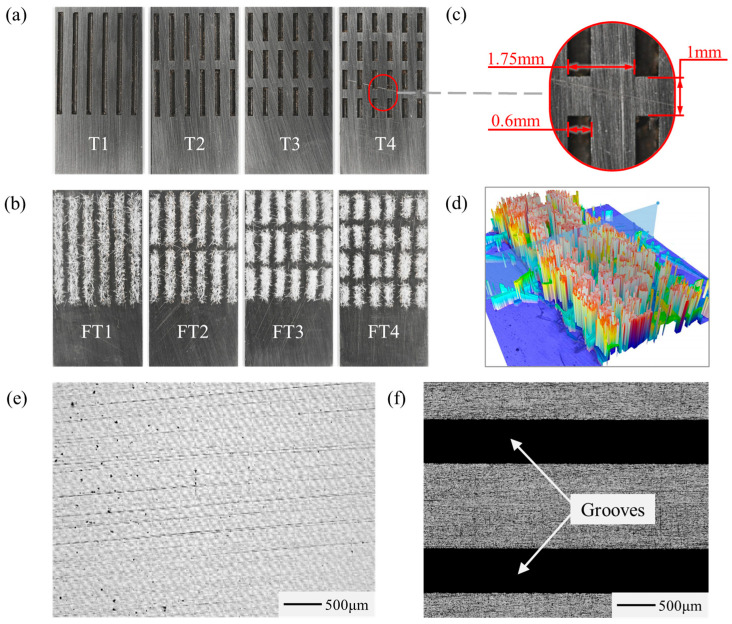
Surface topography of the sample. (**a**) Textured surfaces. (**b**) Flocking surfaces. (**c**) Dimensional parameters of the groove. (**d**) Three-dimensional optical profiler micrograph of the flocking surface. (**e**) Surface before texturing. (**f**) Surface after texturing. (Red arrows) Mark the spacing of the grooves.

**Figure 4 materials-17-04166-f004:**
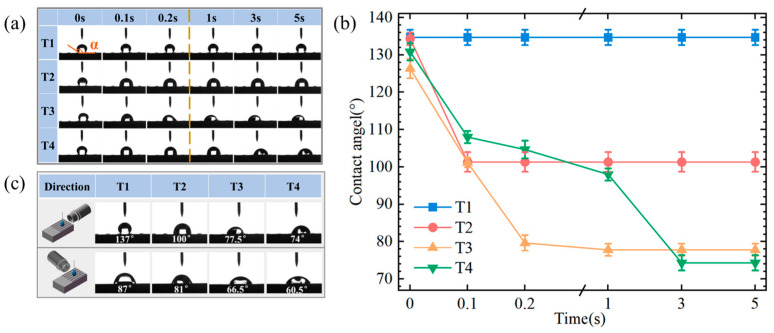
Surface contact angle test. (**a**,**b**) Changes in wetting angle of textured surface with different parameters. (**c**) The spread of droplets on the textured surface in different shooting directions.

**Figure 5 materials-17-04166-f005:**
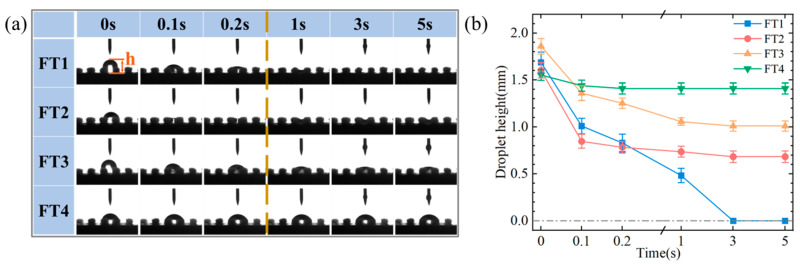
Flocking surface wettability test. (**a**) Wetting characteristics of flocking surfaces with different parameters. (**b**) Variation trends of droplet height.

**Figure 6 materials-17-04166-f006:**
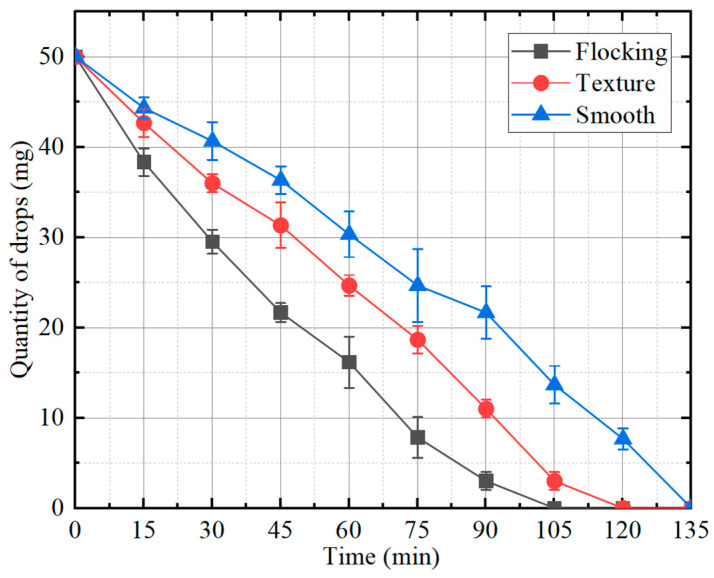
The change in droplet mass with time.

**Figure 7 materials-17-04166-f007:**
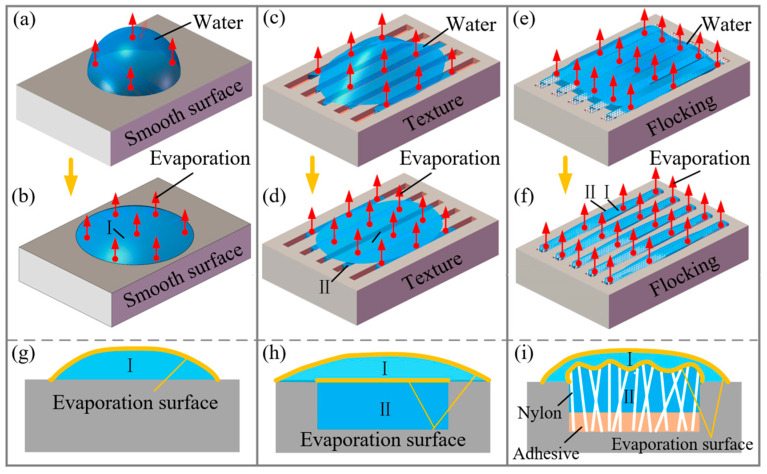
Schematic diagram of water droplets evaporating on the sample surfaces. (**a**,**b**) Evaporation of droplets on a smooth surface. (**c**,**d**) Evaporation of droplets on a textured surface. (**e**,**f**) Evaporation of droplets on a flocking surface. (**g**) Evaporation surface of a smooth surface. (**h**) Evaporation surface of a textured surface. (**i**) Evaporation surface of a flocking surface. (Red arrows) Evaporation of droplets. The number of arrows indicates the rate of evaporation. (Yellow arrows) Droplet evaporation process.

**Figure 8 materials-17-04166-f008:**
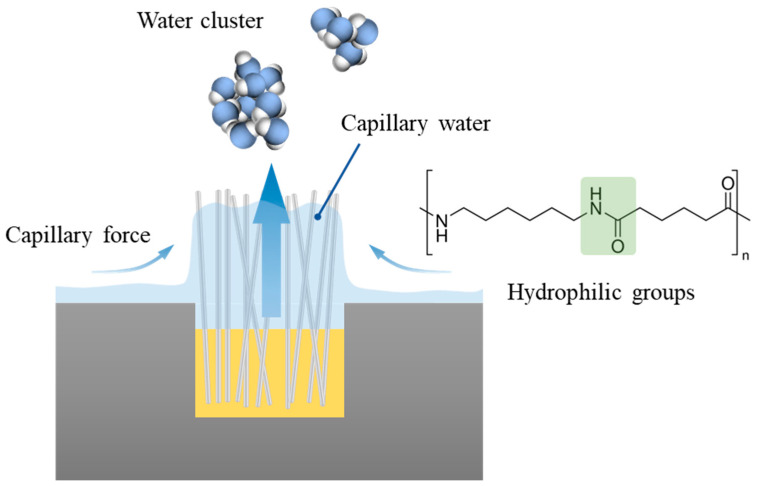
Rapid evaporation mechanism of droplets on flocking surface.

**Figure 9 materials-17-04166-f009:**
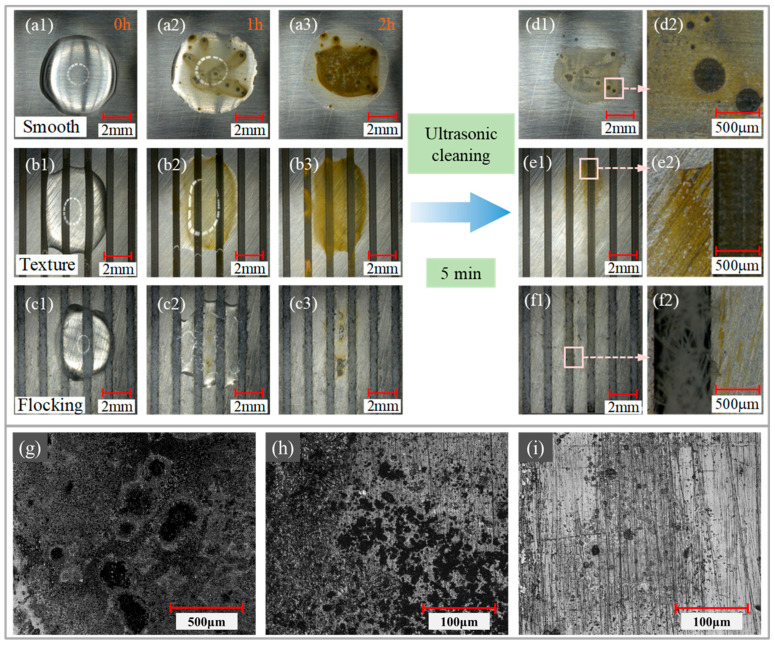
Corrosion process of surfaces under NaCl solution and surface morphology after corrosion. (**a1**–**a3**,**d1**,**d2**) Smooth surface. (**b1**–**b3**,**e1**,**e2**) Textured surface. (**c1**–**c3**,**f1**,**f2**) Flocking surface. (**g**–**i**) Three-dimensional optical profiler images of the smooth surface, textured surface, and flocking surface.

**Figure 10 materials-17-04166-f010:**
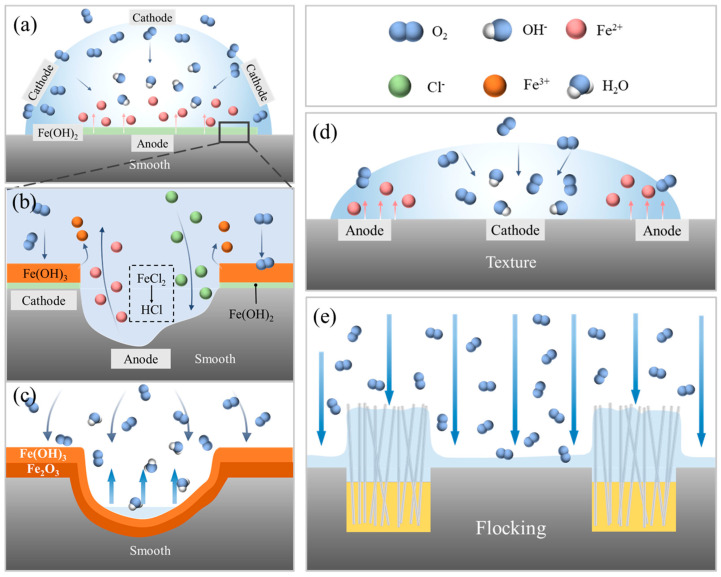
Corrosion process of surfaces. (**a**−**c**) Corrosion process of smooth surface. (**b**) is the enlarged view of the square in (**a**). (**d**) Corrosion process of textured surface. (**e**) Anti-corrosion mechanism of flocking surface.

**Figure 11 materials-17-04166-f011:**
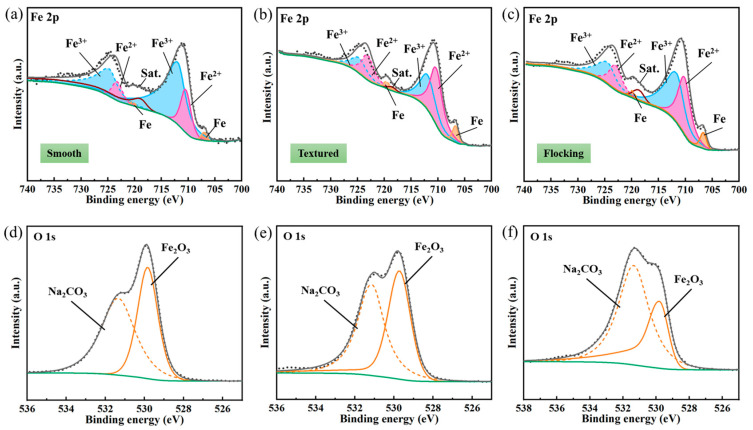
XPS spectra of metal surfaces after corrosion. (**a**–**c**) Fe 2p spectra with split patterns. (**d**–**f**) O 1s spectra with split patterns. (Black lines) High resolution spectrum. (Green lines) Base line.

**Figure 12 materials-17-04166-f012:**
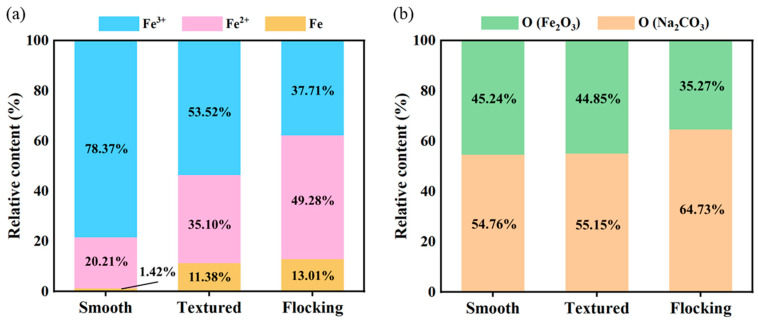
(**a**) Relative content of Fe^3+^, Fe^2+^, and Fe on surfaces. (**b**) Relative content of O of Fe_2_O_3_ and Na_2_CO_3_ on surfaces.

**Table 1 materials-17-04166-t001:** Performance parameters of nylon fiber.

Specific Gravity	Compressive Strength	Bending Strength	Permittivity	Breakdown Voltage	Diameter
g/cm^3^	MPa	MPa		kV/mm	μm
1.15	105	60–100	3.6	16	10

**Table 2 materials-17-04166-t002:** Main elements of M42.

Element	C	Si	Mn	P	S	Cr	V	W	Mo	Co
Content	1.00~1.15	≤0.65	≤0.40	≤0.03	≤0.03	3.50~4.50	0.95~1.35	1.15~1.85	9.0~10.0	7.5–8.5

**Table 3 materials-17-04166-t003:** Parameters of laser system.

Parameter	Value
Power	12 W
Spot diameter	50 μm
Frequency	60 kHz
Processing times	100
Repeat accuracy	±0.003 mm

## Data Availability

The raw data supporting the conclusions of this article will be made available by the authors on request.

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
