# Peer review of "Anti-Corrosion Flocking Surface with Enhanced Wettability and Evaporation"

_materials, 2024, doi:10.3390/ma17164166_

Round 1

Reviewer 1 Report

Comments and Suggestions for Authors

Dear Authors,

The manuscript written very well and introduces a new method to improve corrosion resistance of High-Speed Steel (HSS) surfaces with laser texturing and electrostatic flocking method. The study demonstrates the improvement in wettability, evaporation rates, and anti-corrosion performance. Optimization of groove dimensions and fiber characteristics, with exploring broader applications and long-term durability, could improve the manuscript and its application in different industrial domain. In general, the manuscript is well-prepared and makes a valuable contribution in the field, and I will accept it for publication.

Best regards,

Comments on the Quality of English Language

The manuscript is well-written, with clear and coherent English. It is easy to follow the text. I recommend only for minor language checks.

Author Response

Dear Editor:

Thank you for your suggestions. We have done our best to polish the language in the revised manuscript. The revised manuscript is attached. We hope it will be approved.

Thank you very much for your time and consideration.
Sincerely,
Ni Jing

Reviewer 2 Report

Comments and Suggestions for Authors

This paper deals with the corrosion of tool steel and the use of textured and flocking surfaces as a potential mitigation method. 

There are a few points that need clarifying.

The typical corrosion behaviour of the tool steel should described in the literature review - eg does it suffer from general or localised, pitting type corrosion

In the experimental section, the dimensions of each texture pattern should be given. Depth, width and length. These should be provided for each pattern.

Ideally a micrograph of the textured surface and the flock should be included.

The accuracy of the scales used to measure evaporation should be given.

On line 180, the statement "The optimal parameters for the width, spacing, 180 and depth of the grooves were found to be 0.6 mm, 1.75 mm, and 0.5 mm, respective " It is not clear how was this determined.

A before and after texturing micrograph of the surface should be included or a comment made about the impact of texturing on the near surface layer / microstructure of the steel 

Line 276 discusses evaporation - please state which texture pattern was chosen for these experiments 

Evaporation is performed with one drop, please comment on how you think this relates to a constant flow of liquid. I realise this experiment is beyond the scope but think this point needs acknowledging.

Considering this a tool material, how far does the flocking material protrude from the grooves and what wear resistance does it have? In other words, will the fibres be removed when the tool is operational? 

How densely packed is the flocking fibres - eg g/m2?

When discussing a "smooth" surface, there is no measurement of its Ra. This should be included for completeness 

Figure 6 needs error bars or some comment about repeatability 

Figure 7 appears to show an evaporation surface below the meniscus - please explain how this is possible. 

Figure 9 g,h and i show corrosion. Please say if these were taken within the grooves or on the surface. 

Was the flocking layer removed and images taken underneath? These should be included.

Please say if there any evidence of crevice corrosion under the flocking layer

Line 356 describes pitting corrosion. Pitting depths would add to this paper.

Figure 10 shows an Evans drop but it is incorrect. The anode should be in the centre where the O2 pathway is greatest. The cathode occurs at the edge of the drop where O2 access is easier. 

Line 405 says there is hydrolysis of Cl. Hydrolysis concerns the water molecule and produces H+ to lower the pH. Please clarify. 

There needs to be more images of the corroded surface - from the figure 9 it is appears to show some general corrosion. 

The corrosion discussion would benefit from some electrochemistry testing to show pitting behaviour with and without flocking agent. 

Figure 12 - please say how many areas were tested using XPS and what positions. Were tests performed in or out of the grooves? 

More evidence is needed to prove the flocking agent is providing ant corrosion effects. More micrographs and some electrochemistry would improve this paper and provide more support to claims in the conclusions.

Round 2

Reviewer 2 Report

Comments and Suggestions for Authors

Thank you for addressing the points

Author Response

Dear reviewer:

Thank you very much for your time and consideration.

Sincerely,
Ni Jing